# Cost-effectiveness of advanced life support and prehospital critical care for out-of-hospital cardiac arrest in England: a decision analysis model

Johannes von Vopelius-Feldt, [1,2] Jane Powell,[3] Jonathan Richard Benger[1,2]

This research was presented at the 999 Research Forum in Birmingham, UK, in April 2019.

¹Faculty of Health and Applied Sciences, The University of the West of England, Bristol, UK
²Academic Department of Emergency Care, University Hospitals Bristol NHS Foundation Trust, Bristol, UK
³Centre for Public Health and Wellbeing, University of the West of England, Bristol, Bristol, UK

**Correspondence to**
Dr Johannes von Vopelius-Feldt, Faculty of Health and Applied Sciences, The University of the West of England, Bristol, UK;
johannes.vonvopelius-feldt@uwe.ac.uk

## ABSTRACT

**Objectives** This research aimed to answer the following questions: What are the costs of prehospital advanced life support (ALS) and prehospital critical care for out-of-hospital cardiac arrest (OHCA)? What is the cost-effectiveness of prehospital ALS? What improvement in survival rates from OHCA would prehospital critical care need to achieve in order to be cost-effective?

**Setting** A single National Health Service ambulance service and a charity-funded prehospital critical care service in England.

**Participants** The patient population is adult, non-traumatic OHCA.

**Methods** We combined data from previously published research with data provided by a regional ambulance service and air ambulance charity to create a decision tree model, coupled with a Markov model, of costs and outcomes following OHCA. We compared no treatment for OHCA to the current standard of care of prehospital ALS, and prehospital ALS to prehospital critical care. To reflect the uncertainty in the underlying data, we used probabilistic and two-way sensitivity analyses.

**Results** Costs of prehospital ALS and prehospital critical care were £347 and £1711 per patient, respectively. When costs and outcomes of prehospital, in-hospital and postdischarge phase of OHCA care were combined, prehospital ALS was estimated to be cost-effective at £11 407/quality-adjusted life year. In order to be cost-effective in addition to ALS, prehospital critical care for OHCA would need to achieve a minimally economically important difference (MEID) in survival to hospital discharge of 3%–5%.

**Conclusion** This is the first economic analysis to address the question of cost-effectiveness of prehospital critical care following OHCA. While costs of either prehospital ALS and/or critical care per patient with OHCA are relatively low, significant costs are incurred during hospital treatment and after discharge in patients who survive. Knowledge of the MEID for prehospital critical care can guide future research in this field.

**Trial registration number** ISRCTN18375201

## Strengths and limitations of this study

► This is the first study to describe the cost-effectiveness of the complete care pathway for out-of-hospital cardiac arrest in England.
► This study uses a combination of primary and secondary data, therefore optimising the use of currently available evidence.
► The decision analysis model includes a combination of probabilistic and two-way sensitivity analyses to address the complex question of cost-effectiveness of prehospital critical care in a transparent manner.
► A limitation of this study is that is based on theoretic modelling only; the uncertainty in the underlying data is addressed by the methods and reflected in the presentation of results.

## INTRODUCTION

Rates of survival following out-of-hospital cardiac arrest (OHCA) remain low in the UK, with an overall survival rate to hospital discharge of 7.9%.[1] One area of potential improvement is the provision of prehospital care by emergency medical services (EMS).[2] The current standard of prehospital care in the UK is the provision of advanced life support (ALS) by appropriately trained paramedics.[3] In an attempt to improve survival rates following OHCA, prehospital critical care teams (CCTs) are dispatched to OHCA in some regions in the UK, in addition to the standard ALS response.[4 5] CCTs consist of specialist paramedics and/or doctors, with extended skills and competences, which we previously described in detail.[6] They are frequently based on air ambulances, with funding received variably through charitable donations and/or the National Health Service (NHS).[7]

The cost-effectiveness of ALS care provided by paramedics or physicians has been studied previously.[8 9] However, generalisability of this evidence to current UK practice is limited due to the elapsed time since the research was undertaken and considerable differences in EMS configurations. The cost-effectiveness of prehospital critical care for OHCA has

not been researched in sufficient detail to guide funding decisions. A recently published systematic review of the effectiveness of CCTs for OHCA showed a potential small-to-moderate survival benefit but also identified issues limiting internal and external validities of the included observational studies.[10]

Economic analyses have been undertaken in regard to the cost-effectiveness of helicopter-based emergency medical services care (HEMS) for trauma, with mixed results.[11][12] To our knowledge, no economic analysis exists that addresses the question of cost-effectiveness of prehospital critical care for OHCA. In addition, a recent study of stakeholders' views of prehospital care for OHCA revealed contrasting opinion of whether current NHS funding of prehospital ALS care for OHCA represents an appropriate distribution of limited healthcare resources in the UK.[13] Adding to the complexity of cost-effectiveness of prehospital ALS or critical care for OHCA is the fact that the majority of costs occur along the patient care pathway in hospital or postdischarge.[14][15]

In the absence of robust data regarding the effectiveness of prehospital ALS or critical care for OHCA in the UK setting, this study addresses the following questions:

► What are the costs of prehospital ALS and prehospital care for OHCA in the UK?
► What is the cost-effectiveness of the current OHCA treatment pathway from prehospital ALS care to post-discharge costs, when compared with no treatment?
► What improvement in survival rates from OHCA would prehospital critical care need to achieve in order to be cost-effective, when compared with ALS?

## METHODS
We combined data from previously published research with costs provided by a regional EMS provider and air ambulance charity to create a decision analysis model.

## Development of the decision analysis model
The initial decision analysis model for this research considered three potential outcomes: prehospital death, in-hospital death or survival to hospital discharge. After a focused review of the literature, it became clear that differences in in-hospital treatment costs and survival with either poor or favourable neurological outcome had considerable impact on the cost-effectiveness estimates.[14][15] We therefore expanded the model to include these important branches in a combination of decision tree model for prehospital and hospital costs and outcomes, and a Markov model for postdischarge costs and outcomes. The structure is similar to the model used by Gates et al[15] in their economic analysis of mechanical chest compression during prehospital care for OHCA. Figure 1 shows the final model (for simplicity, only the ALS arm of the model is displayed). Based on literature referenced in the sections below, important assumptions are that

► A proportion of patients survive to hospital arrival but receive palliative care only, resulting in death in the emergency department or shortly afterwards.
► Patients who survive to hospital discharge with Cerebral Performance Category (CPC) scores of 3–4 have a higher rate of survival following the first 5 years after hospital discharge, compared with those patients surviving with CPC scores of 1–2.
► After 5 years postdischarge, annual survival rates for patients still alive at this point are the same as those of the average population.

## Perspective and time horizon
In keeping with the recommendations of the National Institute for Health and Care Excellence (NICE), the perspective on costs chosen for this economic evaluation

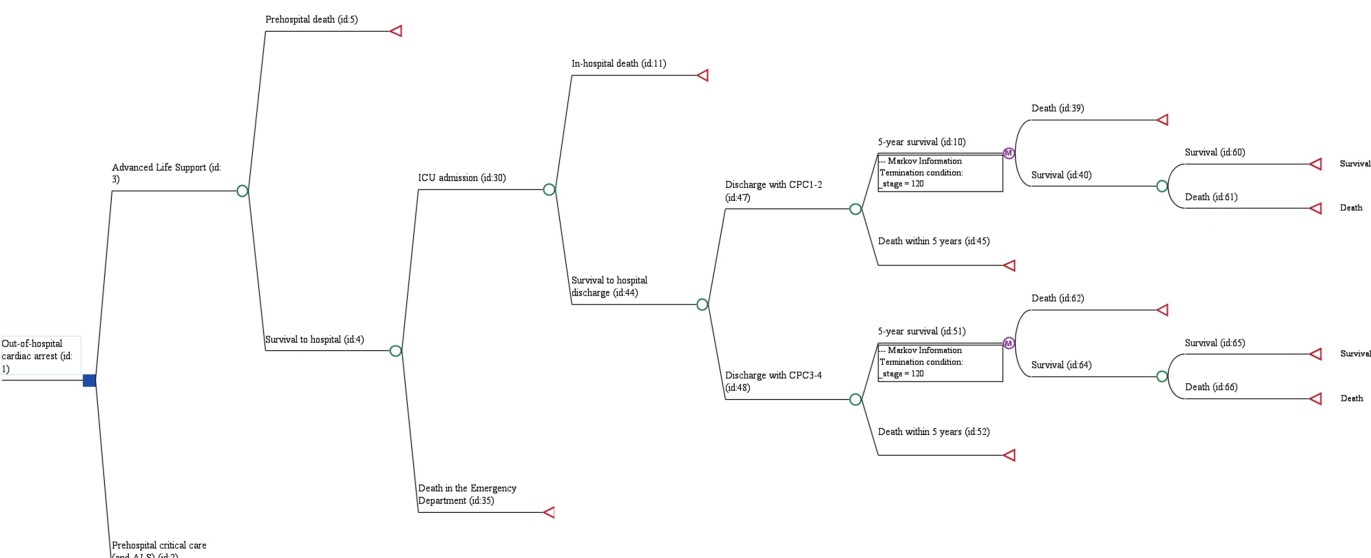

**Figure 1** Final analytic model of prehospital critical care following out-of-hospital cardiac arrest, combining decision tree and Markov model. Due to limited space, the full pathway is shown for the ALS option only. ALS, advanced life support; CPC, Cerebral Performance Category; ICU, intensive care unit.

von Vopelius-Feldt J, et al. BMJ Open 2019;9:e028574. doi:10.1136/bmjopen-2018-028574

was that of the NHS and personal social services.[16] The perspective on outcomes included all direct health effects on patients, namely, length and quality of life (quality-adjusted life years (QALYs)) with a corresponding long-term time horizon.

## Clinical setting

The economic analysis was undertaken in the section of an NHS ambulance trust in England that was covered by a CCT. It covers an area of 2.930 square miles with a population of approximately 2.5 million people. The area is largely suburban or rural with one metropolitan area and two urban areas. The overall population density is 860/ square mile. The patient population is adult, non-traumatic OHCA.

## Alternative interventions: ALS and prehospital critical care

The NHS ambulance trust provided ALS paramedic care, using a combination of first responders, rapid response vehicles and ambulance. The section of the ambulance trust included in this analysis contained 27 ambulance stations with a total of approximately 800 ALS paramedics (employed part time or full time), as well as emergency care assistants and ambulance technicians (data provided by ambulance trust). The median response time for OHCA during the study period was 7.4 min (IQR 4.9–11.4 min), and care on scene was provided by a mean of 2.43 core resources (paramedic-staffed double-crewed ambulances and/or rapid response vehicles). The prehospital critical care service was funded largely through an air ambulance charity but also received support from the ambulance trust. It provided prehospital critical care cover from 07:00–13:00, with access to a helicopter and rapid response vehicles. The team consisted of specialist paramedics in critical care and prehospital critical care doctors. The service responded to OHCAs in addition to the usual ambulance trust's ALS response.

## Identification and synthesis of evidence
### Costs

Financial data were obtained from the ambulance trust and air ambulance charity for the financial year 2015/2016. This included pay for clinical and non-clinical staff, as well as costs of dispatch infrastructure, ambulance stations, vehicles, fuel, medical supplies, communications equipment, community first responders and third-party costs. The amount of funding spent on OHCA was calculated as the number of OHCA cases divided by the number of cases of any aetiology. Calculating the proportion of resources spent on OHCA in this fashion distributes the costs of waiting time of resources in between emergency proportionally between OHCA cases and those of other aetiologies.[17] The costs of in-hospital and postdischarge treatment were synthesised from relevant recent publications, identified by a focused systematic search of the literature (online supplementary appendix 1).[14 15 18]

## Outcomes

After a literature review (online supplementary appendix 2), the following outcomes of interest were included in the decision analysis model (figure 1): survival to arrival at hospital; survival to intensive care unit (ICU) admission, survival to hospital discharge, CPC score at discharge from hospital, 5-year survival following hospital discharge, annual mortality and quality of life. Prehospital outcomes after OHCA with ALS care were based on a 1-year data sample provided by the participating ambulance trust (2016), while in-hospital outcomes were based on recent literature of OHCA in the UK.[19–21] For prehospital critical care, a range of effect sizes were simulated in two-way sensitivity analyses.[10] This approach was chosen to transparently reflect the considerable uncertainty of the effectiveness of prehospital critical care for OHCA in current research.[10] Postdischarge outcomes for the first 5 years after hospital discharge were retrieved from publications identified through a further focused systematic search of the literature (online supplementary appendix 2).[15 22] Further long-term outcomes (quality of life) were modelled according to survival rates provided by the UK's Office for National Statistics.[23 24]

## Application of values and distributions to the model

Prior to inclusion in the model, all costs were adjusted to pound sterling for the financial year 2016–2017 according to the UK Hospital and Community Health Services pay and prices index.[25] In keeping with NICE guidelines, an annual discount rate of 3.5% was applied to costs and effects that occurred after the first year following discharge from hospital.[16] Table 1 provides a comprehensive list of all model parameters based on the sources of data described in the previous section. The CIs or IQRs of the data were transferred to sampling distributions.[26] In keeping with previous recommendations, we used gamma distributions for costs and beta distributions for probabilities.[26] Each parameter in table 1 corresponds to either a cost, effect or probability of each chance node or Markov node of the decision analysis model in figure 1. For costs and effects, these are simply added up for each potential outcome. For example, in-hospital death after prehospital critical care would incur the costs of prehospital critical care plus ALS, costs of ED treatment, and ICU and non-ICU costs of non-survivors. The effect is 0f QALYs, as the outcome is death. The probabilities for each node were entered into the model as the values described in table 1, with the exception of the probability of survival to hospital discharge. To create a 10% rate of survival to hospital discharge after prehospital critical care, for example, the probability value for the corresponding node would need to be entered as a function of '10%, divided by p ICU admission, divided by p survival to hospital'.

## Sensitivity analyses

As the analysis was based largely on secondary data, we undertook a probabilistic sensitivity analysis of the

**Table 1** Values and distributions of parameters of the decision tree and Markov model, all costs adjusted for inflation to the financial year 2016–2017

| Parameters | Cohort/patient group | Mean value (95% CI) | Distribution | Source |
|---|---|---|---|---|
| **Prehospital (decision tree model)** | | | | |
| Costs of prehospital care | ALS | £347 | Gamma | Own data |
| | Critical care (in addition to ALS) | £2058 (£1711 critical care plus £347 ALS care) | N/A | |
| P survival to hospital | ALS | 0.26 (0.25 to 0.27) | Beta | von Vopelius-Feldt et al[10] |
| | Critical care | 0.25 to 0.31 | N/A | |
| **In-hospital (decision tree model)** | | | | |
| Costs of ED treatment | | £377 (£355 to £399) | Gamma | Department of Health[37] |
| P ICU admission* | | 0.65 (0.58 to 0.71) | Beta | Benger et al[19] |
| Costs of ICU treatment (daily) and length of stay on ICU (LOS) | Survival to hospital discharge | £1745 (£1,654 to £1,836) LOS (days) 5.7 (5.4 to 6.0) | Gamma | Petrie et al[14] Nolan et al[20] |
| | In-hospital death | £1768 (£1668 to £1868) LOS (days) 2.4 (2.3 to 2.5) | | |
| Cost of non-ICU treatment (total) | Survival to hospital discharge | £12 823 (£10 123 to £15 523) | Gamma | Petrie et al[14] Nolan et al[20] |
| | In-hospital death | £3835 (£3045 to £4625) | | |
| P survival to hospital discharge† | ALS | 0.09 (0.08 to 0.10) | Beta | von Vopelius-Feldt et al[10] |
| | Critical care | 0.10 to 0.15 | N/A | |
| **Postdischarge (decision tree model)** | | | | |
| P survival with CPC score 1–2‡ | | 0.85 (0.826 to 0.871) | Beta | Phelps et al[22] |
| P 5-year survival | CPC score 1–2 | 0.740 (0.709 to 0.768) | Beta | Phelps et al[22] |
| | CPC score 3–4 | 0.408 (0.332 to 0.489) | | |
| Person-years per death within 5 years | CPC score 1–2 | 1.827 (1.750 to 1.878) | Gamma | Phelps et al[22] |
| | CPC score 3–4 | 0.313 (0.254 to 0.375) | | |
| Utility | CPC score 1–2 | 0.75 (0.70 to 0.80) | Beta | Gates et al[15] |
| | CPC score 3–4 | 0.47 (0.42 to 0.52) | | |
| Annual healthcare costs | CPC score 1–2 | £3358 | Gamma | Gates et al[15] |
| | CPC score 3–4 | £43 670 | | |
| **Long-term outcomes (Markov model)** | | | | |
| P annual mortality after 5 years postdischarge | CPC score 1–2 | National age-adjusted mortality rates | Exact values | Office for National Statistics[23] and Andrew et al[24] |
| | CPC score 3–4 | | | |

*Of survivors to hospital.
†Of all cases of out-of-hospital cardiac arrest.
‡Of survivors to hospital discharge.
ALS, advanced life support; CPC, Cerebral Performance Category; ED, emergency department; ICU, intensive care unit; LOS, length of stay; N/A, not applicable; P, probability.

decision analysis model to reflect the underlying uncertainty in the data.[26] We ran 1000 iterations of the decision analysis model in a Monte Carlo simulation. For each iteration, the values of all model parameters were drawn at random from the sampling distributions described in table 1. In addition, the probabilities of survival to hospital admission and survival to hospital discharge after prehospital critical care were subject to a two-way sensitivity analysis, reflecting a range of potential effects based on a recent systematic review.[10] All analysis was undertaken in TreeAge V.2017.

## Model validation

Validity of decision analysis models can be examined in a number of different ways.[26] Face validity was examined by presenting the model, values and results to a group of prehospital clinicians and air ambulance charity staff. Internal validity was assessed through the sensitivity analyses described previously, as well as selecting extreme values for each individual parameter in the model and checking if the results were influenced as expected. We also undertook visual inspection of the values selected by the software for each variable during the probabilistic sensitivity to ensure that values were in the ranges expected for the parameter. While the software produced an error message if probabilities either did not add up to one or had individual values of less than 0 or more than 1, we also hand checked all probabilities to assure that the overall probability of death/survival was correctly calculated from the probabilities at each step of the decision tree model.

## Presentation of results

In keeping with the three research questions outlined in the introduction, the results will be presented in three sections. First, we will describe the costs occurring during prehospital care for OHCA, for both ALS and critical care. Outcome measure will be pound sterling. We will then describe the cost-effectiveness of prehospital ALS, including the complete patient pathway, when compared with no treatment. The outcome measure will be the incremental cost-effectiveness in pound sterling per QALY. Finally, we will analyse the MEID that prehospital critical care would need to achieve in order to be cost effective at a willingness-to-pay (WTP) threshold of £20 000/QALY. The outcome measure will be the absolute difference in survival to hospital discharge between ALS and prehospital critical care, which fulfils the MEID criteria.

## Patient and public involvement

A patient and public involvement (PPI) group was consulted during the application and planning process of the doctoral research fellowship, which underlies this research. In addition, the PPI groups' views, as well as other relevant stakeholders' views, expressed during previous research[13] informed the methods of this research. Results will be disseminated to all stakeholders who participated, through an executive summary of this research.

**Table 2** Prehospital treatment costs for OHCA

|  | ALS care | Critical care |
|---|---|---|
| Total expenditure 2015–2016 (in £) | | |
| Staff costs | 36 732 844 | 665 293 |
| Vehicles (including fuel) | 8 145 245 | 13 659 |
| Helicopter (including fuel) | – | 1 318 432 |
| Buildings | 2 705 442 | 37 693 |
| Equipment | 1 681 383 | 63 296 |
| Other | 6 261 901 | 37 752 |
| Dispatch centre | 2 226 955 | 45 000 |
| Total | 57 753 770 | 2 181 125 |
| Expenditure per OHCA 2015–2016 (in £) | | |
| Percentage (%) of resources devoted to OHCA | 0.85 | 30.0 |
| Number of patients with OHCA | 1412 | 388 |
| Cost per patient with OHCA | **347** | 1711 |

ALS, advanced life support; OHCA, out-of-hospital cardiac arrest.

## RESULTS

### Prehospital costs

Table 2 gives an overview of the costs of prehospital ALS care and prehospital critical care for OHCA. The total expenditure for ALS care is over 25 times higher than the costs of providing prehospital critical care for the same geographical area. However, OHCAs represent approximately one-third of the workload of the prehospital CCT, whereas only 0.85% of ALS care funding can be attributed to OHCA care. The average cost of a patient with OHCA being attended by ALS paramedics is £347, while the cost for attendance of a prehospital CCT is £1711 (plus £347, as ALS resources would still be dispatched with CCT attendance).

### Cost-effectiveness of ALS for OHCA

The estimated cost-effectiveness of the current standard pathway for OHCA, including prehospital ALS, hospital admission, ICU and non-ICU treatment, as well as post-discharge healthcare costs, is £11 407/QALY (median, IQR £6840/QALY–£16 863/QALY), when compared with no treatment. Figure 2 shows the corresponding scatter plot with results from 1000 iterations of the probabilistic sensitivity analysis. As the majority of these 1000 results lie below the WTP threshold of £20 000/QALY (approximately 93%), prehospital ALS for OHCA can be assumed to be cost-effective at this WTP threshold with a high degree of certainty.

### Incremental cost-effectiveness of prehospital critical care for OHCA

Figure 3 shows the cost-effectiveness acceptability curves for a number of potential treatment effects of prehospital

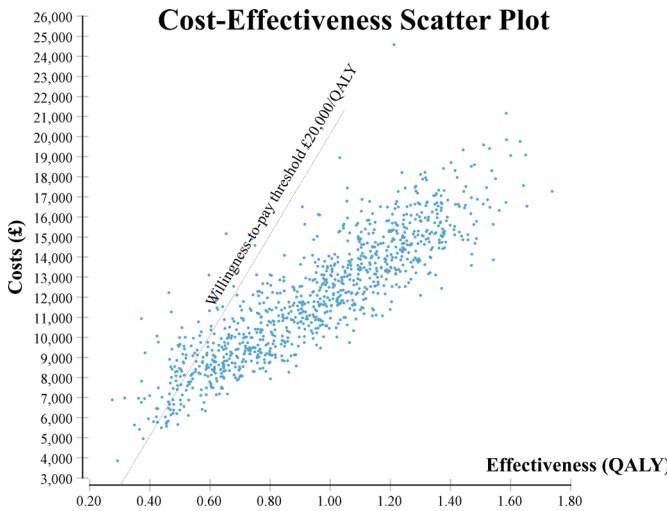

**Figure 2** Scatter plot of the cost-effectiveness of advanced life support for out-of-hospital cardiac arrest, compared with no treatment. QALY, quality-adjusted life year.

critical care following OHCA. For this model, the ratio of rates of survival to arrival at hospital and survival to hospital discharge in the prehospital critical care treatment arm was kept constant. Rates of survival to hospital arrival and of survival to hospital discharge after ALS care for OHCA were 25.8% and 9.0%, respectively. The intersections of the WTP threshold and the cost–acceptability curves provide the probability of prehospital critical care being cost-effective. Prehospital critical care needs to achieve a 2% or 3% absolute increase in survival to hospital discharge to have a 48% or 72% probability of being cost effective at a WTP threshold of £20 000/QALY, respectively. With a lower absolute difference in survival rates of 1%, prehospital critical care has a less than 1% probability of being cost effective at the same WTP threshold. Higher absolute differences in survival of 4%, 5% and 6% result in probabilities of cost-effectiveness at the WTP threshold of £20 000 of 84%, 90% and 93%, respectively.

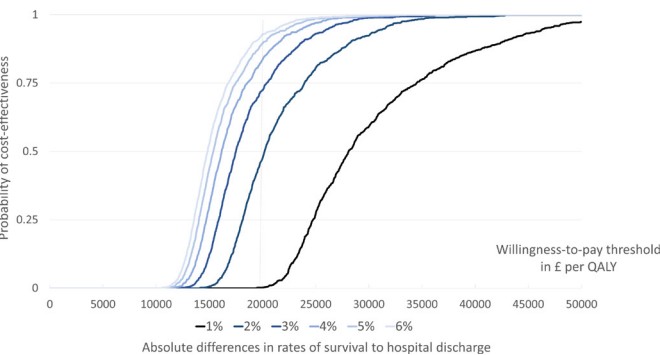

**Figure 3** Cost-effectiveness acceptability curves of different plausible treatment effects of prehospital critical care for out-of-hospital cardiac arrest, when compared with ALS. The underlying baseline survival to hospital discharge for the ALS cohort in this model is 9.0%. ALS, advanced life support; QALY, quality-adjusted life year.

The results of the analysis presented in figure 3 assume that the rates of survival to hospital increase proportionally with the rates of survival to hospital discharge. However, as the majority of costs for the care of OHCA is generated by in-hospital treatments, variations in the rate of survival to hospital can have considerable influence on costs, independent of later survival to hospital discharge. Table 3 provides estimates and IQRs for possible combinations of survival to hospital and survival to hospital discharge rates. As table 3 demonstrates, higher rates of survival to hospital arrival result in moderate increases in incremental costs of prehospital critical care for OHCA.

Table 4 presents a further sensitivity analysis of the effects of varying costs of prehospital critical care and rates of survival to hospital discharge. Variations in the costs of prehospital critical care for OHCA only have a minor-to-moderate effect on its cost-effectiveness.

## DISCUSSION

In this decision model analysis, the cost-effectiveness of prehospital ALS care or prehospital critical care for OHCA was determined to a considerable degree by their effects on short-term and long-term survival, with the majority of costs incurred during hospital treatment or postdischarge.

### Prehospital costs versus later costs

Given the complexity of the economic analysis of the whole OHCA pathway, it is worth first looking at prehospital resource use only. The ambulance service described in this analysis had an annual spending of over £50 million. However, during the study period, patients with OHCA represented only 0.22% of all emergency calls attended. Even with the requirement for more resources per patient with OHCA compared with most other prehospital conditions, only 0.82% of the annual spending was allocated to OHCAs at an estimated cost of £347 per OHCA. This might seem a rather low amount of money to be spent on the immediate care for a life or death situation.[13]

In contrast, the annual costs of providing a helicopter-based prehospital CCT covering the same geographical area was relatively low, at just over 2 million pounds per year. Due to the fact that the CCT attended to much fewer patients (approximately 1200 during the study period) with a higher proportion of OHCAs (30%), the incremental costs of prehospital critical care per OHCA were estimated at £1711. Given the gravity of the acute situation of an OHCA, this incremental cost might be considered an effective use of EMS resources.

Focusing only on prehospital costs of OHCA would make comparison to other healthcare interventions difficult and would ignore the importance of costs accumulating further along the OHCA pathway. For patients with OHCA, where resuscitation is unsuccessful and who are declared dead on scene, no further costs occur. However, a large proportion (25.8% in this cohort) of

**Table 3** Incremental cost-effectiveness of prehospital critical care for out-of-hospital cardiac arrest (in pound sterling per quality-adjusted life year) for different rates of survival to hospital and survival to hospital discharge, compared with advanced life support

| Difference in rates of survival to hospital | Difference in rates of survival to hospital discharge (median, 95% CI) | | | | | |
|---|---|---|---|---|---|---|
| | +1% | +2% | +3% | +4% | +5% | +6% |
| −5% | 25 000 (16 500 to 46 000) | 18 000 (12 500 to 28 500) | 15 500 (11 000 to 23 500) | 14 500 (10 500 to 21 000) | Not achievable | Not achievable |
| 0% | 27 500 (20 000 to 47 500) | 19 000 (14 500 to 30 000) | 16 500 (12 500 to 24 500) | 15 000 (11 500 to 22 000) | 14 000 (11 000 to 20 000) | 13 500 (10 500 to 19 000) |
| +5% | 30 500 (21 500 to 53 000) | 20 500 (15 500 to 33 500) | 17 000 (13 000 to 26 500) | 15 500 (12 000 to 23 000) | 14 500 (11 500 to 21 500) | 14 000 (11 000 to 20 000) |
| +10% | 33 500 (22 500 to 61 000) | 22 000 (16 000 to 37 500) | 18 000 (14 000 to 28 500) | 16 500 (12 500 to 24 500) | 15 000 (12 000 to 22 000) | 14 500 (11 500 to 21 000) |
| +15% | 36 500 (23 000 to 70 500) | 23 500 (17 000 to 42 000) | 20 000 (14 500 to 32 000) | 17 000 (13 000 to 26 500) | 16 000 (12 500 to 23 500) | 15 000 (11 500 to 22 000) |
| +20% | 39 500 (23 500 to 79 500) | 24 500 (17 000 to 46 000) | 20 000 (15 000 to 34 000) | 17 500 (13 500 to 29 000) | 16 500 (12 500 to 25 500) | 15 500 (12 000 to 23 000) |

Light and darker shades of grey indicate most likely combination of differences in rates of survival to hospital and survival to hospital discharge. All costs are in pound sterling and have been rounded to the nearest £500 value.

patients with OHCA survive the prehospital phase of their care and are admitted to a hospital. We estimated the costs of in-hospital treatment to be approximately £22 000 for patients surviving to hospital discharge and £8500 for patients who die in the hospital.[14 20] The major contributors to these costs are ICU-bed days (all patients) and interventions such as primary percutaneous coronary intervention (PPCI), implantable cardioverter defibrillator (ICD) implantation or coronary artery bypass graft surgery (almost exclusively in survivors).[14] Further costs can accumulate after hospital discharge, most importantly the long-term care services required for the small proportion of patients who survive to hospital discharge with poor neurological function (over £40 000/year).[15]

## Cost-effectiveness of ALS for OHCA

We estimated the cost-effectiveness of paramedic-delivered ALS for OHCA to be approximately £11 500/QALY. With the upper limit of the IQR at approximately £16 800, this makes ALS for OHCA almost certainly cost-effective at a NICE WTP threshold of £20 000/QALY.[16]

While a number of previous publications address the cost-effectiveness of individual aspects of prehospital care or hospital care for OHCA,[27–29] only a few address the

**Table 4** Incremental cost-effectiveness of prehospital critical care for out-of-hospital cardiac arrest (in pound sterling per quality-adjusted life year) for a range of costs of prehospital critical care and different rates of survival to hospital discharge, compared with advanced life support

| Difference in costs of prehospital care | Difference in rates of survival to hospital discharge (median, 95% CI) | | | | | |
|---|---|---|---|---|---|---|
| | +1% | +2% | +3% | +4% | +5% | +6% |
| −50% | 20 000 (16 000 to 33 500) | 16 000 (13 000 to 25 500) | 15 000 (12 000 to 23 000) | 14 000 (11 500 to 21 500) | 14 000 (11 000 to 21 000) | 13 500 (11 000 to 20 500) |
| −20% | 25 000 (19 000 to 43,000) | 18 500 (14 500 to 30 500) | 16 500 (13 000 to 26 000) | 15 500 (12 500 to 24 000) | 15 000 (12 000 to 23 000) | 14 500 (11 500 to 22 000) |
| 0% | **28 000 (21 500 to 50 500)** | **20 000 (16 000 to 33 500)** | **17 500 (14 000 to 28 500)** | **16 000 (13 000 to 25 500)** | **15 500 (12 500 to 24 000)** | **15 000 (12 000 to 23 000)** |
| +20% | 31 500 (24 000 to 57 500) | 22 000 (17 000 to 37 000) | 18 500 (14 500 to 30 500) | 17 000 (13 500 to 27 000) | 16 000 (13 000 to 25 500) | 15 500 (12 500 to 24 000) |
| +50% | 36 000 (27 500 to 67 500) | 24,00 (18 500 to 41 500) | 20 000 (16 000 to 33 500) | 18 000 (14 500 to 30 000) | 17 000 (13 500 to 27 000) | 16 000 (13 000 to 25 500) |

All costs are in pound sterling and have been rounded to the nearest £500 value.
Boldfaced values represent the baseline analysis; light grey shaded areas represent likely national variations in costs.

cost-effectiveness of ALS for OHCA in the context of the complete patient pathway. Næss and Steen[30] estimated the cost-effectiveness of ALS for OHCA in Norway to be approximately £6000/QALY in 2004. This much lower estimate is likely due to inflation since the year of publication but also development of in-hospital cardiac arrest care with higher rates of interventions in modern OHCA care and the costs associated with this.[20 31] More recently, Ginsberg *et al*[32] reported the cost-effectiveness of ALS for OHCA to be $28 864 per disability-adjusted life year averted in an Israeli EMS, corresponding to approximately £15 000/QALY–£28 500/QALY.[33] Despite some uncertainty of the data underlying this analysis and taking previous research findings into account, we can be fairly certain that paramedic-delivered ALS is a cost-effective treatment for OHCA in the UK. In addition, in terms of pound sterling per QALY, it compares favourably with a range of interventions currently funded by the NHS.[34 35]

### Costs of prehospital critical care

As the current observational research describing the effectiveness of prehospital critical care for OHCA is inconsistent and likely suffers from confounding,[10] we decided not to include it in this probabilistic sensitivity analysis. Instead, we used a range of potential absolute effect sizes on rates of survival to hospital discharge, ranging from 1% to 6%, reflecting the most likely range of effect sizes.[10] This sensitivity analysis also allowed us to determine a minimally economically important difference (MEID) in survival rates, which would need to be achieved by prehospital critical care services in order to be cost-effective.

At a WTP threshold of £20 000/QALY, the MEID is in the range of 3%–5% absolute improvement in survival to hospital discharge after OHCA. Any improvement in survival rates after OHCA of less than 3%, caused by prehospital critical care services, is very unlikely to be considered cost-effective within an NHS setting.[16] The exact value for the MEID depends on how certain stakeholders want to be about the cost-effectiveness estimate at a given WTP (figure 3), the rates of survival to hospital admission (table 3) and the exact costs of a given prehospital critical care service (table 4). The MEID can be used to guide sample size calculations of future research and to frame the expectations or aims of stakeholders and prehospital critical care services providing care for OHCA.[13] Interestingly, the MEID of prehospital critical care for OHCA calculated in this analysis equals the minimally clinically important difference in survival following OHCA described in previous research.[36]

### Limitations

This economic analysis is based on a theoretical construct of the costs and effects of the care pathway for OHCA. As such, assumptions about events occurring in reality had to be made and had to be assumed to reflect reality accurately, in order for the model to be internally valid.[26] On the other hand, using a modelling approach allowed us to incorporate a considerable number of data sources and to

undertake a range of sensitivity analyses, which improved generalisability. While we undertook steps to ensure the face validity and the internal validity of the model, we were unable to reliably test the external validity (through simulation of real life events).

We did not adapt a microcosting approach that would have, for example, included the exact number of paramedics at the scene for an OHCA or the amount of drugs or equipment used. Such an approach would have likely resulted in an overall more accurate estimate of the costs of ALS care for OHCA, but would have required tremendous efforts to collect and analyse the data. As costs occurring in the later stages of care for OHCA were much higher than those during the prehospital phase, a microcosting approach would have been unlikely to change the cost estimate for prehospital care significantly. Reassuringly, the estimated cost of £347 for ALS care for OHCA compares realistically to the NHS National Schedule of Reference Costs of an average of £236 for 'see and treat and convey' ambulance activities (considering the higher resource use for patients with OHCA).[37]

The main driver of costs for prehospital critical care was the helicopter, followed by staffing costs. We are aware of a number of prehospital critical care services that attend OHCAs using only ground vehicles. This would certainly reduce overall costs of the service (and is therefore reflected in the sensitivity analysis) but also the number of OHCAs that the CCT could reach, particularly in rural/suburban areas. In regard to the validity of the overall model, there are few previous publications to compare to. Reassuringly, Delgado *et al*[38] used a similar decision model analysis approach to examine the potential cost-effectiveness of HEMS for trauma cases in the USA. They found that a 3.7% absolute decrease in mortality was required for HEMS to be cost-effective at a WTP of $50 000 (£38 000)/QALY.

This analysis focuses exclusively on adult, non-traumatic OHCA, as OHCA due to trauma or occurring in the paediatric population is much rarer and due to distinctively different pathology. Results should not be extrapolated to these situations.

### CONCLUSION

This is the first economic analysis, based on a decision analysis model, to address the question of cost-effectiveness of prehospital critical care following OHCA. While costs of either prehospital ALS or critical care per patient with OHCA are relatively low, significant costs are incurred during hospital treatment and after discharge in patients who survive. Taking the whole OHCA care pathway into account, prehospital ALS, as currently delivered in most NHS ambulance trusts, is very likely to be cost-effective at a WTP of £20 000/QALY. For prehospital critical care to be cost-effective at the same WTP threshold, it would need to improve survival from OHCA by approximately 3%–5%.

**Acknowledgements** We thank the South Western Ambulance Service NHS Foundation Trust and the Great Western Air Ambulance Charity for providing the data for this research. We are particularly grateful to the patient and public

involvement group and the other participants in our previous research whose views informed this research.

**Contributors**  JVVF designed the study protocol, collected and analysed the data, and prepared the first draft of the manuscript. JP reviewed the study protocol, supervised the data analysis, and reviewed and revised the manuscript. JRB supervised the research project and reviewed and revised the manuscript.

**Funding**  This work is funded by a National Institute for Health Research (NIHR) doctoral research fellowship for Johannes von Vopelius-Feldt (DRF-2015-08-040). The funder is not involved in the design of the study or collection, analysis and interpretation of data, or in writing the manuscript. The views expressed are those of the authors and are not necessarily those of the NHS, the NIHR or the Department of Health.

**Competing interests**  JVVF and JRB work as prehospital doctors with a regional prehospital critical care team.

**Patient consent for publication**  Not required.

**Ethics approval**  With the exception of the financial data of prehospital care costs, all data were obtained from previously published research or government documents. Therefore, patient consent was not required. The research was reviewed and approved by the Sheffield National Research Ethics Committee, York and Humber on 29 July 2016, reference number 16/YH/0300.

**Provenance and peer review**  Not commissioned; externally peer reviewed.

**Data availability statement**  Data are available upon reasonable request.

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
