## [Reviewer comments · BMJ Open]

ARTICLE DETAILS

TITLE (PROVISIONAL)	Cost-effectiveness of Advanced Life Support and prehospital critical care for out-of-hospital cardiac arrest in England: A decision analysis model
AUTHORS	von Vopelius-Feldt, Johannes; Powell, Jane; Benger, Jonathan

VERSION 1 - REVIEW

REVIEWER	Joshua Reynolds, MD, MS Michigan State University College of Human Medicine United States
REVIEW RETURNED	15-Jan-2019

GENERAL COMMENTS	The authors conducted a cost-effectiveness analysis of prehospital ALS/critical care for OHCA, trying to tease out differences between ALS vs. critical care, and estimating the requisite improvement in clinical outcomes from critical care over and above ALS to keep prehospital critical care cost-effective. Admittedly, I am not a CEA content expert, but I do have some familiarity with the methodology. This appears to be a quality CEA study with satisfactory methodology. The sources of data for modeling have face validity (e.g. NHS data, air ambulance service, other published literature regarding inpatient and post-discharge costs, etc.) and the authors appropriately model a range of values around their point estimates of cost. Likewise, the authors very clearly lay out the Markov model through each successive clinical outcome after prehospital care, and utilize published data to estimate the probabilities of each node in the Markov model. They also model a range of values around their point estimates of clinical outcomes and effectiveness. The two-way sensitivity analyses are especially helpful to the reader and attest to the scope of modeling and sensitivity analyses conducted. Likewise, the manuscript is well-organized and easy to comprehend. I would imagine that a novice reader unfamiliar with CEA studies could follow along and learn from reading this manuscript. Again, outstanding work. I enjoyed reading this submission immensely. I have few one substantive comment for this manuscript:
---

	1) Discussion, page 12, lines 24-28: The authors have conducted excellent work in modeling a complex clinical pathway and timeline. My one critique lies with the assumption that only survivors to hospital discharge accrue the costs of PPCI, ICD, or CABG. Does it prohibitively complicate the model to model a small percentage (e.g. 5%) of non-survivors receiving the higher inpatient costs associated with PPCI, CABG, etc.? Or does it have a minuscule impact on the resulting pound sterling/QUALY.
--	---

REVIEWER	Minaz Mawani The Aga Khan University, Karachi Pakistan
REVIEW RETURNED	25-Jan-2019

GENERAL COMMENTS	1) Research question or objective clearly defined? The title and abstract mention "cost-effectiveness of Advanced Life Support or Prehospital Critical Care for out-of-hospital cardiac arrest...". It should be 'and' instead of 'or' in the title because technically speaking cost-effectiveness of both the interventions have been looked at in this paper. Same is the case with the abstract and the objective part should be modified as "what are the costs of prehospital advanced life support (ALS) and prehospital critical care for out-of-hospital cardiac arrest (OHCA)?....." 5) Regarding research ethics, there needs to be a mention of how ethical considerations were taken care of such as confidentiality of the data, institutional permissions or consent in case of obtaining data on patient outcomes. 12) In the strengths and limitations section mentioned on page 3, kindly elaborate on 2nd and 4th point. 15)The paper would need a language review before getting published with focus on grammar, spelling errors and capitalization of letter when required. The title of figure 2 needs to be revised in order to bring clarity. "cost-effectiveness acceptability curves of difference in treatment effects of prehospital critical care for OHCA as compared to ALS. Baseline survival to hospital discharge for the ALS cohort is 9.0%"
---

REVIEWER	Jordan King University of Utah United States of America
REVIEW RETURNED	25-Mar-2019

GENERAL COMMENTS	The authors developed a decision analysis model to study the costs and cost-effectiveness of ALS and prehospital critical care for out-of-hospital cardiac arrest. As the efficacy of prehospital critical care on survival outcomes is unknown, a primary outcome of the model. The authors found that an improvement in survival of 3-5% with prehospital critical care would be needed for this intervention to be cost-effective. This is an important topic and I commend the authors for their thoughtful efforts. I have a few major and minor concerns: Major Concerns: 1) The only description of the model is a figure which is relegated to the appendix. As this is a de novo model, the development is a
---

	critical aspect of the research. This should be clearly described in the main manuscript including structure and assumptions. 2) I appreciate the authors systematic search for model parameters and the comparison of their findings with other cost-effectiveness analyses of ALS in the discussion section – both work to improve the believability of the model results and are strengths of the study. However, it doesn't appear that efforts were made to validate the predicted outcomes from the model. This is particularly problematic as some of the model estimates (e.g., effect of prehospital critical care on mortality) are unknown, and therefore is a main outcome of the study. At a minimum, the authors should describe efforts to “internally validate” to help the reader feel more comfortable that the model is free of errors. Ideally, the authors would describe efforts to “externally validate” the model. See “Model Transparency and Validation: A Report of the ISPOR-SMDM Modeling Good Research Practices Task Force–7” Minor: 1) There is no mention in the abstract that this is a modeling study. 2) On Table 1, I don't understand the reference of data source as “simulated data”? Please provide a description of what this means. 3) There are several opportunities for a patient to die in the model. Are these probabilities independent? Meaning, are you potentially overcounting death risk? This is an example of where describing your model validation would alleviate concerns. 4) For Table 3, it seems odd to me that cost per QALY goes up as incremental rate of survival to hospital (y-axis) improves? Is this because patients are incurring more costs related to the hospitalization itself? Please describe these results in the manuscript.
--	---

REVIEWER	Emmanuel Drabo Johns Hopkins University, USA
REVIEW RETURNED	02-Apr-2019

GENERAL COMMENTS	This study tackles the very important question of identifying optimal strategies for improving outcomes in out-of-hospital cardiac arrest, with a particular focus on advanced life support (ALS) and prehospital critical care. The authors completed an impressive amount of work to develop a decision-tree/Markov model in order to evaluate these cost and benefit tradeoffs. The study has significant potential to inform and influence the management of out-of-hospital cardiac arrest, if well-articulated. Hence, the comments below are intended to help the authors in that direction. Generally, the presentations of the study objectives, methodological approach, and results are not transparent and are at moment confusing. First, the abstract lacks critical information on the study (e.g., interventions and comparators, perspective, time horizon, outcomes and measures). The expression “decision analysis model” is vague; the authors should be specific and state that they are using a decision-tree model coupled with a Markov model.
--

	In the main analysis, it is unclear what the interventions are and what is the reference (status-quo) intervention. In order to talk about the “cost-effectiveness” of prehospital ALS, one needs a comparator; what is it? Is it no-treatment? There is also a lack of transparency in the methodology used to calculate the costs and QALYs. Paragraphs on how the different elements of the cost and effectiveness measures are combined to produce the total cost and QALY metrics (e.g. how do the different parameters in Table 1 enter the model in Appendix 1 and how are they combined?) would greatly enhance the paper, improve transparency, and allow the study to be reproducible. This is particularly important given that the underlying model, developed in TreeAge, is not provided to readers. In the results section, the authors seem to be making a confusion between the average cost and the incremental cost. The £1,711 cost estimate is certainly not the incremental cost of critical care, if the data in Table 2 are correct: “incremental” relative to what? The paragraph “Cost-effectiveness of Advanced Life Support for OHCA” also needs to be corrected accordingly. One way to improve the presentation of the results is to include a base case analysis result table with strategies and comparators as rows, and the columns being the costs, differences in costs (relative to status quo), QALYs, differences in QALYs (i.e. relative to status quo), and ICERs (relative to status-quo) The authors have a lot of data and information in the Tables and Figures, but do not discuss the key messages of these exhibits in the text of the paper. The paper could be greatly enhanced by summarizing the key messages of each exhibit, when making reference to it. For example, “Figure 1 depicts the results from the probabilistic sensitivity analysis and suggests that xyz.” In Table 1, “Sensitivity analysis” in the Distribution column is misleading; I would recommend leaving blank, since no specific distribution is used. Typically, we plot the incremental cost and effects, so I do not understand how Figure 1 depicts the efficiency of ALS. The notion of efficiency in CEA requires a comparator, which is lacking here. The y-axis of Figure 1 needs to be properly labeled; please remove the legend, as it is confusing. The paper could finally greatly benefit from a careful review for spelling (e.g. “QALY” instead of “QUALY”) and grammar errors.
--	---

VERSION 1 – AUTHOR RESPONSE

Reviewer 1

1) Discussion, page 12, lines 24-28: The authors have conducted excellent work in modelling a complex clinical pathway and timeline. My one critique lies with the assumption that only survivors to hospital discharge accrue the costs of PPCI, ICD, or CABG. Does it prohibitively complicate the model to model a small percentage (e.g. 5%) of non-survivors receiving the higher inpatient costs

associated with PPCI, CABG, etc.? Or does it have a minuscule impact on the resulting pound sterling/QUALY.

Thank you for your comment. The estimate of in-hospital costs is based on a detailed analysis by Petrie et al. (reference 14), and actually includes a small number of non-survivors receiving a few of the above interventions. We have corrected the corresponding section in the manuscript to the accurate description of

'The major contributors to these costs are ICU-bed days (all patients) and interventions such as PPCI, ICD implantation or coronary artery bypass graft surgery (almost exclusively in survivors).[14]'

Reviewer 2

1) The title and abstract mention "cost-effectiveness of Advanced Life Support or Prehospital Critical Care for out-of-hospital cardiac arrest...". It should be 'and' instead of 'or' in the title because technically speaking cost-effectiveness of both the interventions have been looked at in this paper.

Thank you, we have changed the title accordingly.

2) Same is the case with the abstract and the objective part should be modified as "what are the costs of prehospital advanced life support (ALS) and prehospital critical care for out-of-hospital cardiac arrest (OHCA)?....."

We have also updated the abstract text, as suggested.

3) Regarding research ethics, there needs to be a mention of how ethical considerations were taken care of such as confidentiality of the data, institutional permissions or consent in case of obtaining data on patient outcomes.

Apologies for the oversight of not including an ethics statement. Other than financial data, all data related to outcomes and treatments was taken from previously published research or government documents, so no confidentiality issues arose. We added the following paragraph to the end of the methods section

'Ethics and consent

With the exception of the financial data of prehospital care costs, all data were obtained from previously published research or government documents. Therefore, patient consent was not required. The research was reviewed and approved by the Sheffield National Research Ethics Committee, York and Humber on 29 July 2016, reference number 16/YH/0300.'

4) In the strengths and limitations section mentioned on page 3, kindly elaborate on 2nd and 4th point.

The BMJ Open guidelines only allow for one sentence per bullet point in this section. We have tried to clarify point 2 and 4 as much as possible within these constraints

- 'This study uses a combination of primary and secondary data, therefore optimising the use of currently available evidence'
- 'A limitation of this study is that is based on theoretic modelling only; the uncertainty in the underlying data is addressed by the methods and reflected in the presentation of results'

5) The paper would need a language review before getting published with focus on grammar, spelling errors and capitalization of letter when required.

Thank you. The revised manuscript has been reviewed by the two co-authors, both native English speakers.

6) The title of figure 2 needs to be revised in order to bring clarity. "cost-effectiveness acceptability curves of difference in treatment effects of prehospital critical care for OHCA as compared to ALS. Baseline survival to hospital discharge for the ALS cohort is 9.0%"

See below, we have tried to clarify the title.

'Figure 2. Cost-effectiveness acceptability curves of different plausible treatment effects of prehospital critical care for OHCA, when compared to ALS. The underlying baseline survival to hospital discharge for the ALS cohort in this model is 9.0%.'

Reviewer 3

1) The only description of the model is a figure which is relegated to the appendix. As this is a de novo model, the development is a critical aspect of the research. This should be clearly described in the main manuscript including structure and assumptions.

Thank you for your comment. We have included a short description of the process developing the model and moved the figure into the main text. We also attached a copy of the TreeAge file to the revised manuscript, which readers will be able to access.

'Development of the decision analysis model

The initial decision analysis model for this research considered three potential outcomes: prehospital death, in-hospital death or survival to hospital discharge. After a focused review of the literature, it became clear that differences in in-hospital treatment costs and survival with either poor or favourable neurological outcome have considerable impact on the cost-effectiveness estimates.[14, 15] We therefore expanded the model to include these important branches in a combination of decision tree model for prehospital and hospital costs and outcomes, and a Markov model for post-discharge costs and outcomes. Figure 1 shows the final model used (for simplicity, only the ALS arm is displayed). Important assumptions, based on literature referenced below, are that a proportion of patients survive to hospital arrival but only receive palliative care, resulting in death in the emergency department or shortly afterwards. Patients who survive to hospital discharge with cerebral performance category

(CPC) of 3-4 have a higher rate of survival following the first 5 years after hospital discharge, compared to those patients surviving with CPC 1-2. After 5 years post-discharge, annual survival rates for patients still alive at this point are the same as those of the average population.'

2) I appreciate the authors systematic search for model parameters and the comparison of their findings with other cost-effectiveness analyses of ALS in the discussion section – both work to improve the believability of the model results and are strengths of the study. However, it doesn't appear that efforts were made to validate the predicted outcomes from the model. This is particularly problematic as some of the model estimates (e.g., effect of prehospital critical care on mortality) are unknown, and therefore is a main outcome of the study. At a minimum, the authors should describe efforts to "internally validate" to help the reader feel more comfortable that the model is free of errors. Ideally, the authors would describe efforts to "externally validate" the model.

Thank you for this comment, we have added the mechanism undertaken for face and internal validity to the methods section. In regards to external validation, due to the scarcity of previous research on the subject, we struggled to find data which we could simulate, which wasn't already part of the model. We have added this to the limitations section.

'Model validation

Validity of decision analysis models can be examined in a number of different ways.[26] Face validity was examined by presenting the model, values and results to a group of prehospital clinicians and air ambulance charity staff. Internal validity was assessed through the sensitivity analyses described above, as well as selecting extreme values for each individual parameter in the model and checking if the results were influenced as expected. We also undertook visual inspection of the values selected by the software for each variable during the probabilistic sensitivity, to assure that values were in the ranges expected for the parameter. While the software produced an error message if probabilities either did not add up to 1 or had individual values of less than 0 or more than 1, we also hand checked all probabilities to assure that the overall probability of death / survival was correctly calculated from the probabilities at each step of the decision tree model.'

'Limitations

[...] While we undertook steps to assure face validity and internal validity of the model, we are unable to reliable test external validity (through simulation of real life events).'

3) There is no mention in the abstract that this is a modeling study.

We have expanded the abstract, also in keeping with reviewer 4's comments.

4) On Table 1, I don't understand the reference of data source as "simulated data"? Please provide a description of what this means.

Apologies, this referred to the fact that the values for these parameters were 'simulated' in a one- or two-way sensitivity analyses. As the values were chosen based on our systematic review, we have replaced 'simulated data' with a reference to the systematic review.

5) There are several opportunities for a patient to die in the model. Are these probabilities independent? Meaning, are you potentially overcounting death risk? This is an example of where describing your model validation would alleviate concerns.

While the software produces an error message if probabilities either don't add up to 1 or have individual values of less than 0 or more than 1, we also hand checked the probabilities to assure that the overall probability of death / survival is correctly calculated from the probabilities at each step of the decision tree model. We added this to the methods section, see reply to comment 3.

6) For Table 3, it seems odd to me that cost per QALY goes up as incremental rate of survival to hospital (y-axis) improves? Is this because patients are incurring more costs related to the hospitalization itself? Please describe these results in the manuscript.

This effect is indeed due to the considerably larger costs of in-hospital treatment, when compared to prehospital care, and the reason it is included in the sensitivity analysis. We have added an explanation of these findings.

'The results of the analysis presented in figure 3 assume that the rates of survival to hospital increase proportionally with the rates of survival to hospital discharge. However, as the majority of costs for the care of OHCA is generated by in-hospital treatments, variations in the rate of survival to hospital can have considerable influence on costs, independently of later survival to hospital discharge. Table 3 provides estimates and interquartile ranges for possible combinations of survival to hospital and survival to hospital discharge rates. As table 3 demonstrates, higher rates of survival to hospital arrival result in moderate increases in incremental costs of prehospital critical care for OHCA.'

Reviewer 4

1) First, the abstract lacks critical information on the study (e.g., interventions and comparators, perspective, time horizon, outcomes and measures). The expression "decision analysis model" is vague; the authors should be specific and state that they are using a decision-tree model coupled with a Markov model.

Thank you for the guidance on improving the clarity of the abstract and manuscript. We have updated the abstract to include a clearer description of intervention/comparators and the decision model. Due to the constraints of the word limit (297 of 300 currently), we had to leave the description of perspective, time horizon, outcomes and measures to the methods section.

'Methods

We combined data from previously published research with data provided by a regional ambulance service and air ambulance charity to create a decision tree model, coupled with a Markov model, of costs and outcomes following OHCA. We compared no treatment for OHCA to the current standard of care of prehospital ALS, and prehospital ALS to prehospital critical care. To reflect the uncertainty in the underlying data, we used probabilistic and two-way sensitivity analyses.'

2) In the main analysis, it is unclear what the interventions are and what is the reference (status-quo) intervention. In order to talk about the “cost-effectiveness” of prehospital ALS, one needs a comparator; what is it? Is it no-treatment?

Thank you for pointing this out, we were indeed not clear about the comparator of ALS care (the usefulness of describing it only really became clear during the study process). We have now clarified in the abstract (see reply to comment 1) and in the methods section that ‘no treatment’ is the comparator for ALS care. Ideally, this would be a BLS prehospital system, however, as these don’t exist anymore in the UK and ALS is the current standard of care, we chose to pragmatically work with what the data available, to provide a general idea of current healthcare costs for OHCA. We added the following paragraph to the methods section

‘Presentation of results

In keeping with the three research questions outlined in the introduction, the results will be presented in three sections. First, we will describe the costs occurring during prehospital care for OHCA, for both ALS and critical care. Outcome measure will be Pound Sterling (£). We will then describe the cost-effectiveness of prehospital ALS, including the complete patient pathway, when compared to no treatment. The outcome measure will be the incremental cost-effectiveness in £/QALY. Finally, we will analyse the MEID which prehospital critical care would need to achieve in order to be cost effective at a willingness-to-pay (WTP) threshold of £20,000/QALY. The outcome measure will be the absolute difference in survival to hospital discharge between ALS and prehospital critical care which fulfils the MEID criteria.’

3) There is also a lack of transparency in the methodology used to calculate the costs and QALYs. Paragraphs on how the different elements of the cost and effectiveness measures are combined to produce the total cost and QALY metrics (e.g. how do the different parameters in Table 1 enter the model in Appendix 1 and how are they combined?) would greatly enhance the paper, improve transparency, and allow the study to be reproducible. This particularly important given that the underlying model, developed in TreeAge, is not provided to readers.

Thank you for this comment. We updated the methods section to describe this in more detail. We tried to upload the TreeAge file for readers to access but the system only allows for PDF files to be uploaded as supplementary materials. We have attached the file ‘for editors only’ to this submission and included a data sharing agreement which states that readers can request the TreeAge file from the first author.

‘Each parameter in table 1 corresponds to either a cost, effect or probability of each chance node or Markov node of the decision analysis model in figure 1. For costs and effects, these are simply added up for each potential outcome. For example, in-hospital death after prehospital critical care would incur the costs of prehospital critical care plus ALS, costs of ED treatment, and ICU and non-ICU costs of non-survivors. The effect is 0 QALYs, as the outcome is death. The probabilities for each node were entered into the model as the values described in table 1, with the exception of the probability of survival to hospital discharge. To create a 10% rate of survival to hospital discharge after prehospital critical care, for example, the probability value for the corresponding node would need to be entered as a function of ‘10%, divided by p ICU admission, divided by p survival to hospital’.

‘DATA AVAILABILITY STATEMENT

The analytic model used for this research (TreeAge file) can be requested by sending an email to the author of this publication.’

4) In the results section, the authors seem to be making a confusion between the average cost and the incremental cost. The £1,711 cost estimate is certainly not the incremental cost of critical care, if the data in Table 2 are correct: “incremental” relative to what? The paragraph “Cost-effectiveness of Advanced Life Support for OHCA” also needs to be corrected accordingly.

1,711 is the additional costs of the attendance of a prehospital critical care team at the scene of OHCA. As ‘incremental costs’ suggest a certain methodology, we deleted this from the re-written paragraph, this paragraph is simply a description of prehospital costs, without any consideration of effectiveness.

‘Table 2 gives an overview of the costs of prehospital ALS care and prehospital critical care for OHCA. The average cost of a patient with OHCA being attended by ALS paramedics is £347, while the cost for attendance of a prehospital critical care team is £1,711 (plus £347, as ALS resources would still be dispatched with critical care team attendance).’

We updated the cost-effectiveness of ALS for OHCA and legend of figure 2 to clarify that the cost-effectiveness described here is compared to no treatment.

‘Cost-effectiveness of Advanced Life Support for OHCA

The estimated cost-effectiveness of the current standard pathway for OHCA, including prehospital ALS, hospital admission, ICU and non-ICU treatment, as well as post-discharge healthcare costs, is £11,407/QALY (median, IQR £6,840 /QALY – £16,863/QALY), when compared to no treatment. Figure 2 shows the corresponding scatterplot with results from 1,000 iterations of the probabilistic sensitivity analysis.’

‘Figure 2. Scatterplot of the cost-effectiveness of Advanced Life Support for out-of-hospital cardiac arrest, compared to no treatment.’

5) One way to improve the presentation of the results is to include a base case analysis result table with strategies and comparators as rows, and the columns being the costs, differences in costs (relative to status quo), QALYs, differences in QALYs (i.e. relative to status quo), and ICERs (relative to status-quo)

Thank you for this suggestion, we agree that this is a useful and common way of presenting such results. We admit that our study is slightly unusual in that we used the methods somewhat in reverse in regards to cost-effectiveness of prehospital critical care, using the minimally economical important difference (MEID). The reason for this approach, as opposed to simply including the results of the systematic review in the model, is the concern about underlying confounding in favour of prehospital critical care. As a result, we would struggle to create the suggested table in a coherent and useful way.

6) The authors have a lot of data and information the Tables and Figures, but do not discuss the key messages of these exhibits in the text of the paper. The paper could be greatly enhanced by summarizing the key messages of each exhibit, when making reference to it. For example, “Figure 1 depicts the results from the probabilistic sensitivity analysis and suggests that xyz.”

We tried to avoid mixing presentation and interpretation of results, but admit that we might have gone a bit too far, leaving the reader unsupported. We have added the following short key messages for each relevant table / figure of the results section, as suggested.

'Table 2 gives an overview of the costs of prehospital ALS care and prehospital critical care for OHCA. The total expenditure for ALS care is over 25 times higher than the costs of providing prehospital critical care for the same geographical area. However, OHCA's represent approximately one third of the work load of the prehospital critical care team, whereas only 0.85% of ALS care funding can be attributed to OHCA care. The average cost of a patient with OHCA being attended by ALS paramedics is £347, while the cost for attendance of a prehospital critical care team is £1,711 (plus £347, as ALS resources would still be dispatched with critical care team attendance).'

'Figure 2 shows the corresponding scatterplot with results from 1,000 iterations of the probabilistic sensitivity analysis. As the majority of these 1,000 results lie below the WTP threshold of £20,000/QALY (approximately 93%), prehospital ALS for OHCA can be assumed to be cost-effective at this WTP threshold with a high degree of certainty.'

'Figure 3 shows the cost-effectiveness acceptability curves for a number of potential treatment effects of prehospital critical care following OHCA. For this model, the ratio of rates of survival to arrival at hospital and survival to hospital discharge in the prehospital critical care treatment arm was kept constant. Rates of survival to hospital arrival and of survival to hospital discharge after ALS care for OHCA were 25.8% and 9.0%, respectively. The intersections of the WTP threshold and the cost-acceptability curves provide the probability of prehospital critical care being cost-effective. Prehospital critical care needs to achieve a 2% or 3% absolute increase in survival to hospital discharge to have a 48% or 72% probability of being cost effective at a WTP threshold of £20,000 per QALY, respectively. With a lower absolute difference in survival rates of 1%, prehospital critical care has a less than 1% probability of being cost effective at the same WTP threshold. Higher absolute differences in survival of 4%, 5% and 6% result in probabilities of cost-effectiveness at the WTP threshold of £20,000 of 84%, 90% and 93%, respectively.'

'Table 3 provides estimates and interquartile ranges for possible combinations of survival to hospital and survival to hospital discharge rates. As table 3 demonstrates, higher rates of survival to hospital arrival result in moderate increases in incremental costs of prehospital critical care for OHCA.'

'Table 4 presents a further sensitivity analysis of the effects of varying costs of prehospital critical care and rates of survival to hospital discharge. Variations in the costs of prehospital critical care for OHCA only have a minor to moderate effect on its cost-effectiveness.'

In Table 1, "Sensitivity analysis" in the Distribution column is misleading; I would recommend leaving blank, since no specific distribution is used.

We have replaced 'sensitivity analysis' with N/A in table 1.

7) Typically, we plot the incremental cost and effects, so I do not understand how Figure 1 depicts the efficiency of ALS. The notion of efficiency in CEA requires a comparator, which is lacking here. The y-axis of Figure 1 needs to be properly labeled; please remove the legend, as it is confusing.

We have now updated the manuscript at various locations, as well as the figure legend, to clarify that the comparator is no treatment in this case. We removed the legend, since we agree it was not providing any useful information. The y-axis labelling was lost in formatting, this has now been added.

8) The paper could finally greatly benefit from a careful review for spelling (e.g. "QALY" instead of "QUALY") and grammar errors.

We replaced all erroneous mentioning of QUALY with QALY, and the revised manuscript has been reviewed by both co-authors who are native English speakers.